*A Nature Portfolio journal*

# YmoA functions as a molecular stress sensor in *Yersinia*
Tifaine Héchard [1,4], Lu Lu[1,4], Tomas Edgren[2], Chi Celestine[3] & Helen Wang [1] ✉

Pathogenic bacteria sense and respond to environmental fluctuations, a capability essential for establishing successful infections. The YmoA/Hha protein family are conserved transcription regulators in Enterobacteriaceae, playing a critical role in these responses. Specifically, YmoA in *Yersinia* adjusts the expression of virulence-associated traits upon temperature shift. Still, the molecular mechanisms transducing environmental signals through YmoA remain elusive. Our study employs nuclear magnetic resonance spectroscopy, biological assays and RNA-seq analysis to elucidate these mechanisms. We demonstrate that YmoA undergoes structural fluctuations and conformational dynamics in response to temperature and osmolarity changes, correlating with changes in plasmid copy number, bacterial fitness and virulence function. Notably, chemical shift analysis identifies key roles of a few specific residues and of the C-terminus region in sensing both temperature and salt-driven switch. These findings demonstrate that YmoA acts as a central stress sensor in *Yersinia*, fine-tuning virulence gene expression and balancing metabolic trade-offs.

Pathogenic bacteria have evolved numerous regulatory mechanisms to modulate virulence and metabolic gene expression in response to environmental cues, such as temperature and osmolarity changes. These mechanisms are essential for successful host invasion[1]. The human pathogenic *Yersinia* relies on a plasmid-encoded Type III secretion system (T3SS) to evade the host immune system during infection. This conserved T3SS is highly regulated and responds to environmental shifts encountered by *Yersinia* and other successful enteric pathogens during infection, such as elevated temperature, pH gradients, osmolarity changes and redox status[2].

The hemolysin expression-modulating protein (Hha), was first identified in 1991 as a thermoregulator and osmotic regulator of α-haemolysin toxin expression in *Escherichia coli*[3]. Proteins of the Hha family serve as pivotal intermediaries in sensing environmental variables and transducing these signals to the DNA-binding global regulator, H-NS (histone-like nucleotide-structuring protein). H-NS has the ability to bind to curved AT-rich DNA commonly found in promoter regions, thus inhibiting the expression of numerous genes in Gram-negative bacteria[4,5]. Hha, Cnu (oriC-binding nucleoid-associated, also called YdgT, a paralog of Hha) and their homologous in *Yersinia*, YmoA (*Yersinia* modulator), exhibit high sequence identity and are functionally interchangeable in *E. coli*[6,7]. Studies have shown that these proteins can sense diverse stress signals, including temperature, acidity and oxygen levels, impacting the expression of virulence-associated traits (Cornelis et al.[7–15] Wallar, Bysice, and Coombes[16]). Thus, the Hha family of proteins plays a critical role in

orchestrating gene expression in response to environmental stress signals, a feature highly conserved across Enterobacteriaceae.

*Yersinia* possess a virulence plasmid, pYV (plasmid of *Yersinia* Virulence), encoding some of its most important virulence factors such as the T3SS, a needle-shaped complex, the Yops (the effector proteins secreted by the T3SS) and the adhesion factor YadA[17,18]. All these genes are expressed at 37 °C upon contact with the host-target cell. They are under the control of *lcrF*, the main transcriptional activator of pYV-encoded genes[19]. YmoA is primarily known to be the thermosensitive repressor of *lcrF*[20]. YmoA is believed to regulate T3SS gene expression through its effect on DNA supercoiling and its interaction with H-NS[21,22]. Moreover, YmoA also plays a role in insecticidal gene regulation at environmental temperature in *Y. enterocolitica* (Starke and Fuchs [23]).

YmoA/Hha complex with H-NS has been extensively characterized[9,22,24,25]. X-ray crystallography and nuclear magnetic resonance (NMR) studies have elucidated their structural interactions, providing insights into their binding interfaces[26,27]. Mutational studies have identified critical residues involved in this interaction and their impact on gene regulation[24].

Recent studies have demonstrated that *Yersinia* possess an additional regulatory mechanism to modulate T3SS function, other than the well-studied LcrF system, based on the virulence plasmid copy number (PCN). This reversible PCN regulation allows adjustment of plasmid-encoded gene dosage to trade off the metabolic burden of the T3SS during infection[28,29].

[1]Department of Medical Biochemistry and Microbiology, Uppsala University, Uppsala, Sweden. [2]Rarity Bioscience, Uppsala, Sweden. [3]Biophysics, Discovery Sciences, Astra Zeneca, Uppsala, Sweden. [4]These authors contributed equally: Tifaine Héchard, Lu Lu. ✉e-mail: helen.wang@imbim.uu.se

In this study, using a combination of biophysical and biochemical assays, we rationalized the role of YmoA in sensing temperature and osmolarity changes and transducing this signal into phenotypical responses. We deepened our understanding of YmoA regulation of gene expression using RNAseq, Western blots, and fitness measurements. Noticeably, we identified YmoA as a key regulator of PCN. PCN is essential for virulence, however it is a novel mechanism and its regulation remains poorly understood. Using NMR spectroscopy analysis, we observed conformational changes of four distinct residues across three domains in helix 1, helix 3, and helix 4 upon temperature shift. These residues interact with H-NS binding sites, suggesting a signal pathway through this interaction. Additionally, a cluster of residues in helix 2 and 4 underwent conformational changes in response to salt titration, indicating their role in the signal transduction via salt-sensing. We identified critical residues able to sense both temperature and osmolarity changes. Mutagenesis of these residues confirmed their importance in YmoA regulation. Our study enhances the understanding the central role of YmoA as a molecular environmental sensor, particularly in temperature and osmolarity.

## Results

### YmoA is a highly conserved transcriptional regulator in Enterobacteriaceae

The YmoA/Hha family exhibits a high degree of conservation, especially within helices 1, 2, and 3, with variations at both termini (Fig. 1a and S1). YmoA protein sequences are identical between *Y. pseudotuberculosis* and *Y. pestis*, sharing 89.2% and 82.2% sequence identity with Hha from *E. coli* (PDB ID: 2k5s), and *S.* Typhimurium (PDB ID: 4icg) respectively [30]. YmoA exhibits a divergence of five amino acids following methionine (Met-1) at the N-terminus when compared to Hha from Enterobacteriaceae family, such as *E. coli, S.* Typhimurium, and *K. pneumoniae* (Fig. 1a). Importantly, despite this N-terminal variation, the binding of H-NS was observed in both complexes, indicating that the missing residues are not crucial for interaction between H-NS and YmoA/Hha[26,27]. Other variations include differences in Glu-36 and Leu-37 in helix 3, and the last four residues in helix 4 across species (Fig. 1a).

Amino acids in equivalent positions typically share similar properties, except for Ser-65 in Hha, replaced by Pro-60 in YmoA, reducing loop flexibility and changing helix 4 orientation. This shifts the C-terminus further from the interface compared to Hha (Fig. 1b, c). YmoA interacts with H-NS through Glu-20, Glu-36, and Asp-43, and the C-terminal helix, while Hha interacts with H-NS through Glu-25, Ala-41, Aps-48, and the C-terminal helix[24,27]. Specifically, Ala-41 of Hha in *E. coli* interacts with Asn-9 of H-NS, whereas in *Yersinia*, YmoA's Glu-36 forms a salt bridge with Asn-9. Structurally, YmoA and Hha are superimposable in helices 1, 2, and 3. However, the C-terminus, which is known to play a critical role in interaction with H-NS, is structurally different in YmoA in *Yersinia* compared to Hha in other species. In summary, both sequential and structural analyses demonstrate a high conservation within YmoA/Hha family, with the differences mainly localized in the C-terminus.

### YmoA controls global gene expression in *Yersinia*

YmoA is known to interact with H-NS, a global regulator of gene expression[4,22]. *ymoA* is part of an operon together with a second gene, named *ymoB*. In *E. coli* and *Salmonella*, *ymoB* homologous, *tomB*, was proposed to be part of a toxin-antitoxin system with Hha. In order to delineate the role of the *ymoBA* operon in regulating gene expression *Y. pseudotuberculosis*, we performed transcriptional analysis and compared the transcriptomes of the of *ymoA* and *ymoB* deletional mutant with that of the wildtype strain. Three independent cultures of *Y. pseudotuberculosis* wildtype and single *ymoA* and *ymoB* deletional mutant were grown at 26 °C to late exponential phase. RNA was isolated and subjected to transcriptomic analysis. It is evident from our RNA-sequencing analysis that YmoA is intricately involved in the regulation of numerous transcripts in the bacterium (Fig. 2a and S2a, b). Notably, a significant portion of the genes down-regulated by YmoA are plasmid-

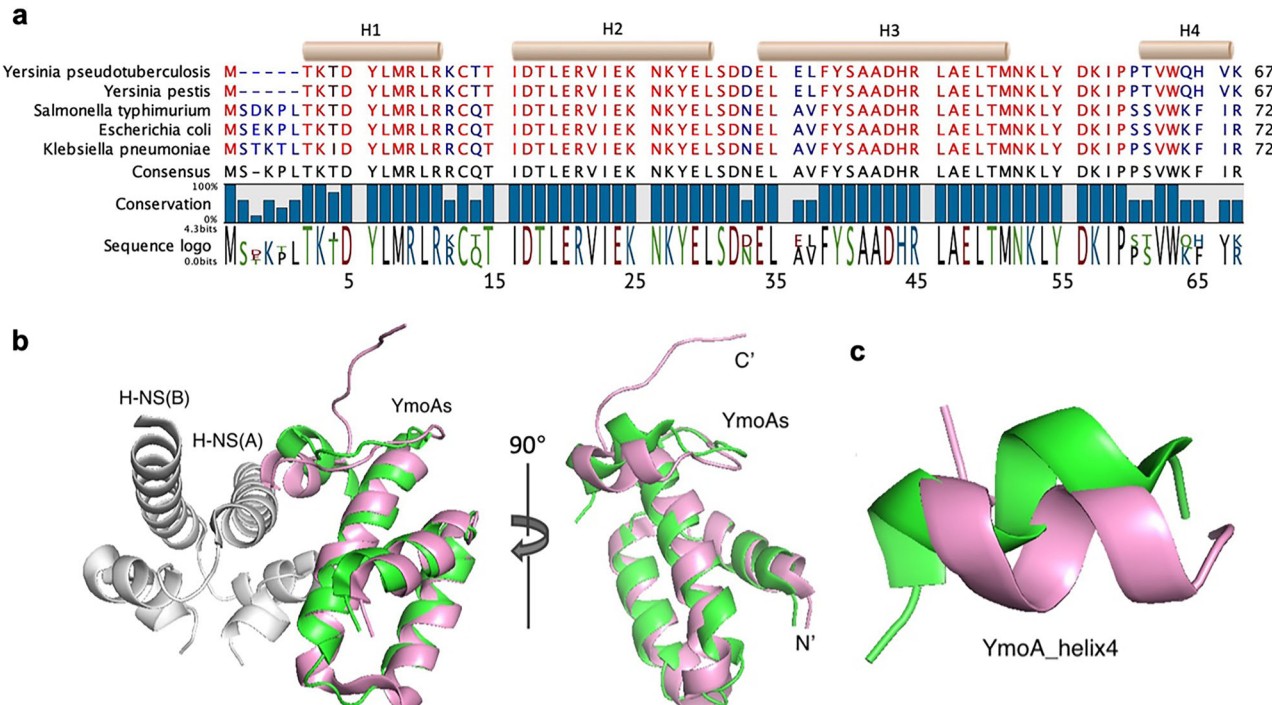

**Fig. 1 | YmoA protein family is generally conserved among species. a** Hha/YmoA sequences were aligned in CLC Main workbench. The coloring scheme from red to blue was based on the conservation score of each amino acid ranging from 50% to 100%. The four helices (top beige bars) were annotated from the secondary structure of Hha from *Salmonella* Typhimurium (PDB ID: 4icg). **b, c** Structural alignment of *Y. pseudotuberculosis* YmoA (PDB ID: 2k5s, pink) overlaid with *E. coli* Hha (PDB ID: 2mw2, green). **b** YmoA and Hha global structure alignment showed that they shared a similar overall fold with RMSD of Cα at 1.63. **c** Local presentation of helix 4 in the global structural alignment showed two helices pivoted apart from each other for roughly 30 degrees.

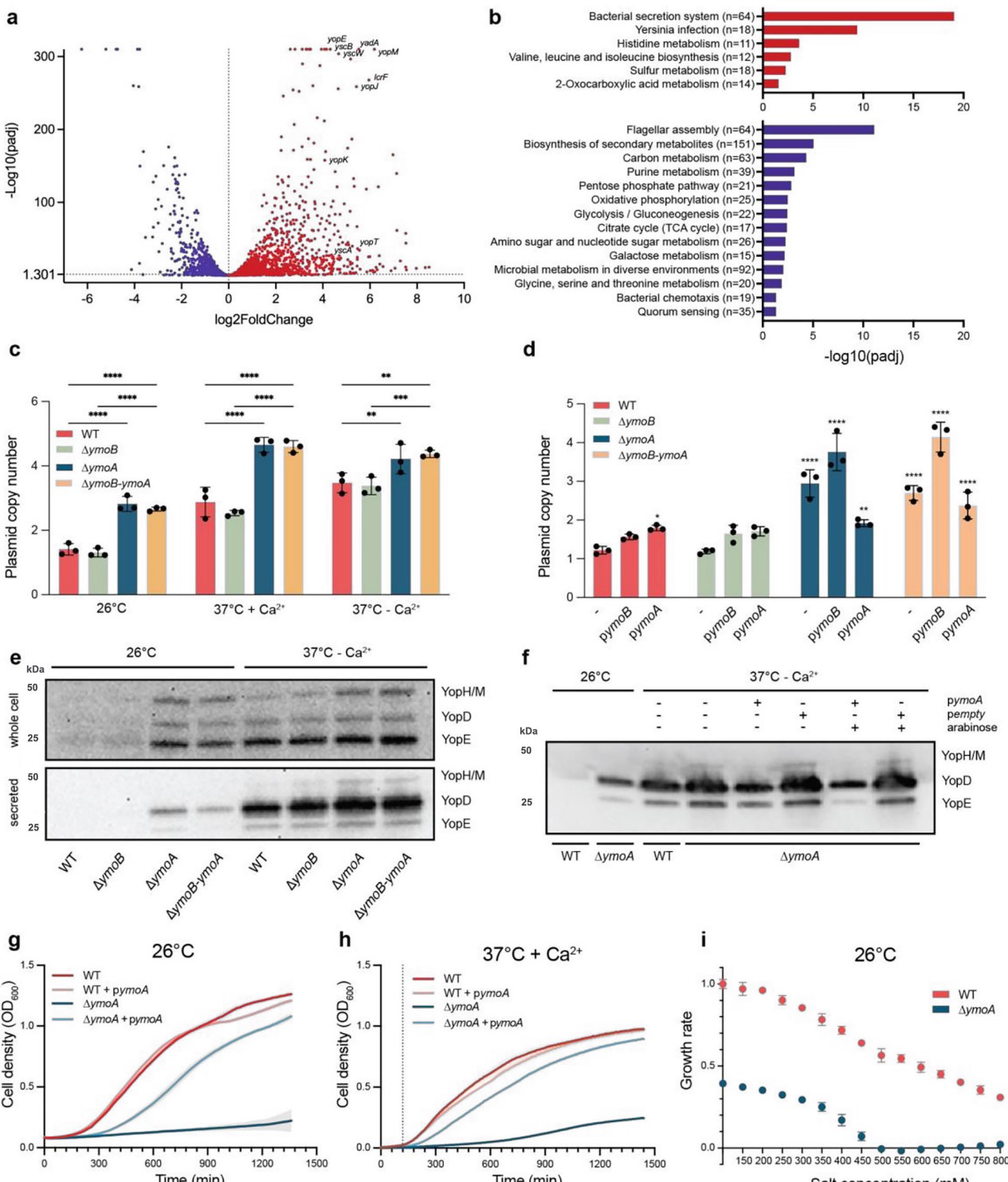

**Fig. 2 | YmoA controls global gene expression in response to environmental changes. a**, **b** YmoA control of gene expression determined by RNAseq. Strains were grown into logarithmic phase at 26 °C for 3 h. The plots show genes that are over-expressed (red) and repressed (purple) in a Δ*ymoA* mutant compared to a wild-type. **c–i** Strains were grown at 26 °C for 2 h and then shifted (**c**, **d,g–i**). to an additional 3 h at 26 °C or at 37 °C with or without Ca$^{2+}$ or (**c–f**). to an additional 1 h at 26 °C or at 37 °C without Ca$^{2+}$. **c**, **d** PCN determined using ddPCR. The data represent means ± SD (*n* = 3) (**P < 0.01; ****P < 0,0001 using (**c**). 2way Anova post ho**c:** Tukey's test comparing the different strains within one condition (**d**). one-way Anova post hoc: Dunett's test). (-) indicates non trans-complemented strains. **e**, **f** Western blots performed on centrifuged pellet (whole cell) or the supernatant (secreted) using anti-Yops/anti-YopE antibodies. Shown is one representative Western Blot from one out of 3 biological replicates. **g–i** Growth determined by measuring the OD$_{600}$. The dotted line represented the time at which the temperature was switched from 26 °C to 37 °C. The data represent means ± SD (*n* = 3).

encoded, encompassing those related to the bacterial T3SS and the Yop effector proteins, pivotal elements for *Yersinia* infection (Fig. 2b and Table S7). It's worth highlighting that YmoA also down-regulates *lcrF* (*virF*) expression, an important temperature-driven transcriptional activator (Fig. S2b). This evidence supports the notion that YmoA plays a crucial role in Yersinia virulence, as the *ymoA* deletion mutant shows a reduced ability to colonize and disseminate in a mouse model[31]. In addition, YmoA exerts a up-regulatory effect on numerous other genes belonging to several function categories, including those involved in flagellar assembly, carbon metabolism and metabolites biosynthesis pathways (Fig. 2b and Table S7). Our transcriptomic analysis clearly showed that YmoA is a global regulator affecting gene expression in *Yersinia*. Interestingly, we found that YmoB also plays a role in the regulation of T3SS and Yops genes but to a lesser extent than YmoA (Fig. S2a, c). YmoB also appears to be involved in ribosomal gene down-regulation.

### YmoA modulates the plasmid copy number and T3SS to trade-off virulence costs associated with infection in response to environmental changes

YmoA is known to repress T3SS expression at 26 °C (Cornelis et al.[8]). Recently, PCN has been identified as a novel mechanism for regulating the dosage of virulence genes in response to temperature[29]. We measured PCN using ddPCR to explore whether YmoA-dependant T3SS repression is associated with PCN regulation in *Y. pseudotuberculosis*. Our findings revealed that both Δ*ymoA* and Δ*ymoB-ymoA* mutants exhibited approximately twice the PCNs compared to wild-type at 26 °C ($p < 0.0001$) (Fig. 2c). At 37 °C + Ca$^{2+}$, the PCN increased from about three copies in the wild-type to five copies in the Δ*ymoA* and Δ*ymoB-ymoA* mutants ($p < 0.0001$). At 37 °C − Ca$^{2+}$, this discrepancy between the wild-type and the mutants was less pronounced yet still significant ($p < 0.01$). Notably, the Δ*ymoB* mutant did not show a significant deviation from the wild-type under any of the three conditions tested. To further verify that the observed PCN changes were attributed to YmoA, we measured PCN in the trans-complemented strains at 26 °C (Fig. 2d). The complementation of YmoA in the Δ*ymoA* mutant effectively reduced the PCN closer to a wild-type levels. Interestingly, a Δ*ymoA* mutant complemented with p*ymoB* displayed an increased PCN compared to the Δ*ymoA* alone ($p = 0,0011$). Taken together, these results suggest that YmoA plays a regulatory role in PCN control.

To further understand the role of YmoA in regulating T3SS activity, we analyzed the expression and secretion of Yops proteins by SDS-PAGE analysis (Fig. 2e, f). At 26 °C, we observed that Yops effector proteins were both slightly expressed and secreted in Δ*ymoA* and Δ*ymoB-ymoA* mutants, in contrast to the wild-type and Δ*ymoB* mutant (Fig. 2e). To ensure that the observed secretion was not a result of cellular lysis, we analyzed various sample fractions by SDS-PAGE followed by silver-staining. No significant changes were observed in background protein level (Fig. S3a). In T3SS inductive conditions, all strains both expressed and secreted Yops. Interestingly, we found an enhanced secretion of Yops in all three mutant strains compared to the wild-type; Δ*ymoB* ($p = 0.0405$), the Δ*ymoA* ($p = 0.0128$) and the Δ*ymoB-ymoA* mutant ($p = 0.0260$) (Fig. 2e and S3b). Conversely, when YmoA was overexpressed, there was a repression in the secretion of Yops compared to the wild-type (Fig. 2f). These findings align with our previous observation that PCN is increased at 26 °C and 37 °C in a Δ*ymoA* mutant. Collectively, the data suggest that YmoA acts as a negative regulator for the T3SS expression and secretion.

YmoA is essential for regulating the expression of the T3SS at 26 °C (Cornelis et al.[8]). Given that expressing and secreting the T3SS apparatus is costly for bacteria, we examined how the absence of YmoA affects the fitness cost of bacteria under different conditions. We found that at 26 °C, the growth of the Δ*ymoA* mutant was severely impaired, exhibiting only 15% of the wild-type strain 's relative growth rate. However, this growth rate was partially rescued to ~75% of the wild-type level when *ymoA* was trans-complemented in the mutant (Fig. 2g). Moreover, overexpression of *ymoA* in the wild-type strain did not result in any growth defects. This observation is likely attributed to the de-regulation of T3SS in the absence of YmoA. No

significant fitness cost was observed in the Δ*ymoB* mutant compared to the wild-type (Fig. S4a) while the double Δ*ymoB-ymoA* mutant displayed a phenotype similar to the Δ*ymoA* mutant (Fig. 5b). At 37 °C in T3SS-repressive conditions, the growth of the Δ*ymoA* mutant remained significantly impaired compared to the wild-type. Similar to what was observed at 26 °C, trans-complementation of YmoA in the Δ*ymoA* mutant rescued most of the fitness cost, bringing it closer to the wild-type level. This suggests that YmoA continues to plays a role in repressing the T3SS even at elevated temperatures (Fig. 2h). Furthermore, we explored the role of YmoA in salt-sensing (Fig. 2i). The wild-type showed a linear decrease in growth rate with increasing salt concentrations. The Δ*ymoA* mutant also demonstrated a slowdown in growth rate with increased salt concentrations, as observed in the wild-type, but the trends differed. Specifically, at 100 mM NaCl, the growth rate of the Δ*ymoA* mutant was reduced by about 60% compared to the wild-type. At 500 mM NaCl, the Δ*ymoA* mutant was unable to grow at all anymore. These results suggest a distinct sensitivity to salt concentrations between the wild-type and the Δ*ymoA* mutant.

In order to determine whether the fitness cost observed in the Δ*ymoA* and Δ*ymoB-ymoA* mutants was exclusively due to the T3SS expression, we cured their virulence plasmid, which carries the T3SS genes, and subsequently measured their growth rate. The loss of the pYV plasmid restored the growth rate of the Δ*ymoA* and Δ*ymoB-ymoA* mutants regaining about 70% of the wild-type growth rate at 26 °C (Fig. S4c), indicating that the plasmid-encoded T3SS largely contributes to the growth reduction at 26 °C in the absence of YmoA.

### YmoA is degraded slowly and still present in the cells 3 h after temperature shift

YmoA is degraded at 37 °C by two proteases, ClpXP and Lon[20]. In an effort to understand the dynamics of YmoA temperature regulation, we investigated its expression level in *Y. pseudotuberculosis* under different conditions by Western blotting. Noticeably, we observed that despite its degradation at 37 °C, YmoA persisted at reduced levels even after 3 h at 37 °C (Fig. S5). Interestingly, the Δ*ymoB* mutant showed lower YmoA levels than the wild-type across all tested conditions. This implies that while YmoA is degraded at 37 °C, a significant amount of the protein remains within the cells for a period following the temperature shift. This observation hints at the possibility of an additional, more rapid mechanism to alleviate YmoA regulation of gene expression at 37 °C.

### YmoA is folded and stable in vitro

Before attributing the observed changes in amino acids flexibility in YmoA to environmental factors, it is crucial to confirm that YmoA is properly folded and stable after cloning and expression in *E. coli*. In *Y. pseudotuberculosis*, YmoA's CD spectra showed a typical alpha helical structure, with minimal variation under various conditions. The CD spectra indicated a gradual loss of secondary structure with a rising temperature, likely due to YmoA unfolding as the temperature increased (Fig. S6a). At the same temperature, YmoA possessed the same composition of the secondary structures, when it was tested with three different salt concentrations, ranging from 37.5 mM to 500 mM NaCl (Fig. S6b). Additionally, thermal shift assay revealed a characteristic denaturation pattern of YmoA. Initially stable with low fluorescent emission at low temperatures, it began denaturation as temperature rose. The melting temperature was ~60 °C, where the inflection point crossed (Fig. S6c), corroborating with CD results that YmoA maintained its structure in conditions used for NMR studies. Furthermore, when plotting far-UV CD signal as a function of temperature, it was also observed that YmoA retained the same secondary struture composition at physiological temperature up to 60 °C (Fig. S6d).

### YmoA undergoes a temperature-dependent conformational change and regulates *Yersinia* growth

Temperature variations from the host equilibrium temperature (~310 K) are known to influence the expression of virulence genes in pathogenic bacteria[32]. The YmoA/H-NS complex regulates T3SS genes expression in

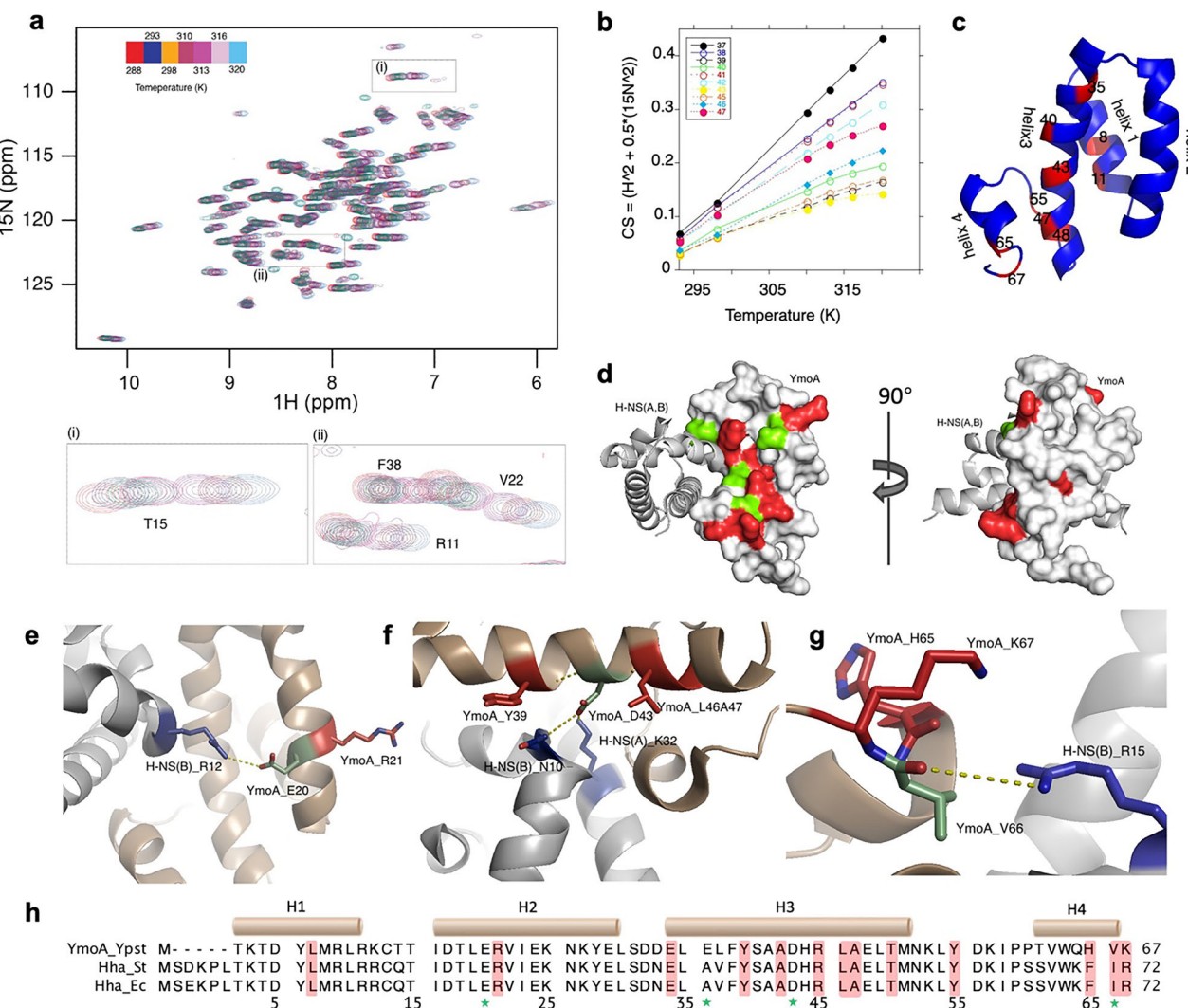

**Fig. 3 | A few thermally sensitive residues are flanking three interacting residues, and form intensive networks. a** NMR $^1$H-$^{15}$N correlation spectroscopy showing the chemical shifts at increasing temperature of YmoA. **b** A plot of chemical shift residues observed to undergo temperature dependent conformational change. Sigmoidal behavior of the fit is seen for these residues. **c** Overall distribution of temperature-sensitive residues (red). **d** Presentation in the surface mode. A cluster of temperature sensing amino acids (Red) points towards the opposite of the YmoA-H-NS interface (interacting residues from YmoA in green). The right ensemble is the left ensemble with a horizontal rotation of 90°. **e–g** The network formed by three interacting residues in YmoA (green): Glu-20; Asp-43; Val-66. **h** The temperature sensing residues were highlighted in red shade in the sequence alignment of YmoA and Hha from *E. coli* and *S.* Typhimurium. Residues with green stars at the bottom are H-NS interacting residues.

response to temperature[20]. To verify whether YmoA participates in temperature-dependent conformational changes in *Y. pseudotuberculosis*, we used NMR spectroscopy to measure amino acid positional shifts upon temperature change. Our study revealed that residues in helices 1, 3, and 4 of YmoA undergo temperature-dependent conformational alterations. Specifically, residues Leu-7, Arg-21, Glu-34, Tyr-39, Ala-42, Arg-45, Leu-46, Ala-47, Thr-50, Tyr-55, His-65, Val-66, and Lys-67 showed sigmoidal shifts in NMR with increasing temperature (Fig. 3a, b), indicating that YmoA undergoes conformational change in response to temperature variations.

Overall, the temperature-sensing residues (marked red) predominantly reside along a surface adjacent to the YmoA/H-NS interface (Fig. 3c, d, h). For a clearer understanding of potential signal pathways, we aligned each helix of YmoA with its counterpart in Hha from *S.* Typhimurium (Fig. 3e–g). Notably, Glu-20 in YmoA, close to the temperature-sensing residue Arg-21, interacts with Arg-12 in chain B of H-NS. The temperature sensor Val-66 directly engages with Arg-15 in chain B of H-NS. Additionally, Asp-43, interacting with Asn-10 in the dimer-forming helix in

chain B of H-NS, and Lys-32 in chain A, forms a complex network crucial for the dimerization of H-NS. The same Asp-43 also interacts with temperature-sensing residues Tys-39 and Ala-47 in adjacent helix turns. Overall, the temperature-sensing residues form a network with residues interacting with H-NS via hydrogen bonds within three amino acids clusters (marked green).

In *E. coli* and *S.* Typhimurium, Ala-41 is known to directly interact with Asn-9 in H-NS[26,27]. However, in *Yersinia*, this position is occupied by a glutamic acid instead of alanine. No direct interaction was found between Glu-36 and temperature-sensing residues in YmoA, although it is spatially close to Glu-34 and Tyr-39 (two of the temperature-sensing residues). Structural alignment shows that YmoA Glu-36 resides on the opposite side of the dimerization interface, facing H-NS. Due to the complex's asymmetry, Glu-36's position varies: it is near Lys-6 in chain B in Hha from *S.* Typhimurium when aligned with chain C, and close to Cys-21 in chain B when aligned with chain D. This suggests that Glu-36 likely interacts with H-NS, but its specific role in signal sensing and transmission remains unclear due to the asymmetric positioning and sequence variation.

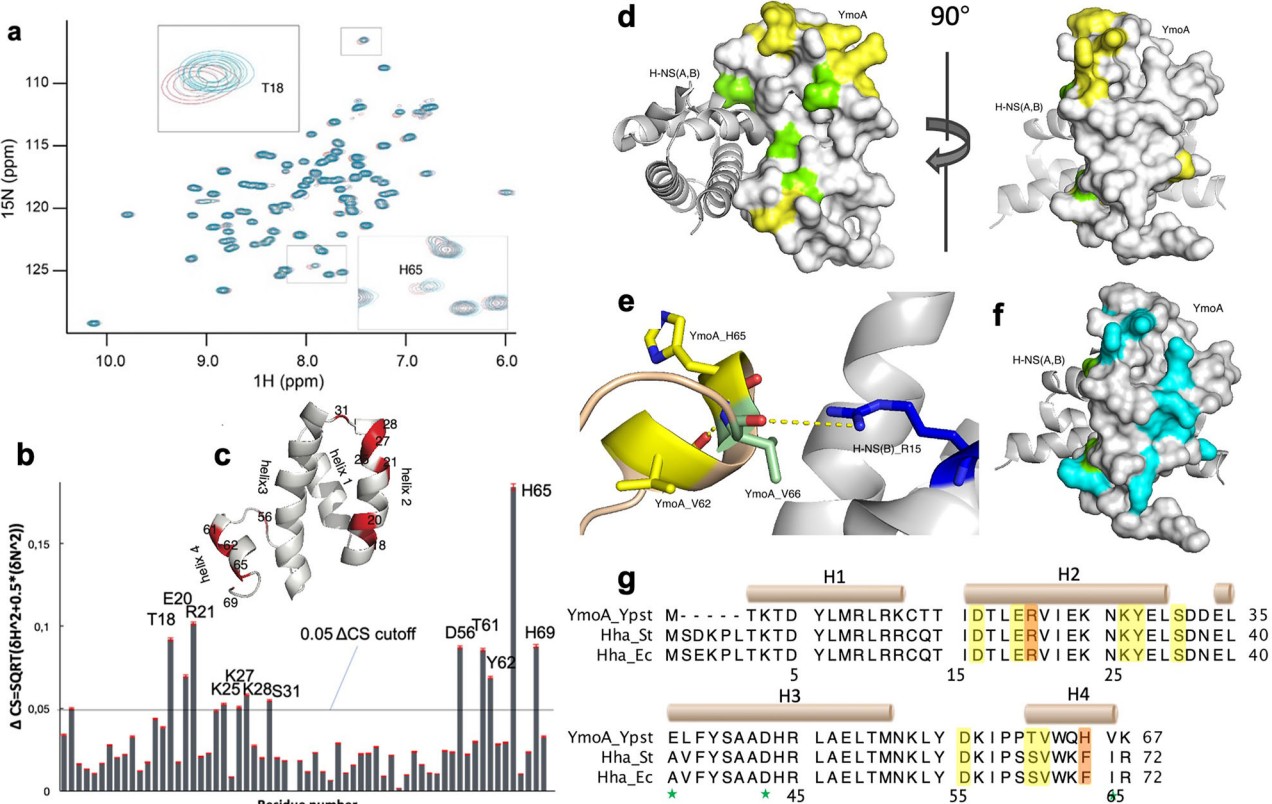

**Fig. 4 | The salt-sensing residues of YmoA are distributed in helix 2 and 4. a** NMR $^1$H-$^{15}$N correlation spectroscopy showing the chemical shifts at increasing salt concentrations of YmoA. **b** A plot of the difference of the chemical shift change at low and high salt with respect to amino acid sequence. The chemical shift (CS) difference was computed according to the following equation (CS = SQRT(($\Delta$N$^2$)/ 5 + $\Delta$H$^2$), with red error bars indicating SD. **c** Overall distribution of salt-sensing residues (red). **d** Presentation in the surface mode. Salt-sensing residues (yellow) were mainly charged residues on the surface, and H-NS interacting residues (green) on the interface. **e** The network formed by Val-66. **f** Overall distribution of all charged residues (cyan) on the surface. **g** The salt sensing residues were highlighted in yellow shade in the sequence alignment of YmoA and Hha from *E. coli* and *S. typhimurium*. Arg-21 and His-65 are both temperature and salt sensitive, and marked in orange shade. Residues with stars at the bottom are H-NS interacting residues.

### YmoA undergoes a long-range conformational relaxation upon binding to salt-specific sites

In order to verify the effect of salt on the conformation and dynamics of YmoA, we performed a salt dependent binding titration of YmoA in *Y. pseudotuberculosis* (Fig. 4). From our salt interaction studies we observed that a few amino acids in YmoA exerted a salt dependent chemical shift change. This type of change often signifies a direct interaction of the amino acids which undergo chemical shift change or the effect of a neighboring amino acids exerting this interaction. Alternatively, it may also be due to a long-range conformational effect. Here, we observed that residues in helix 2 and 4 experience a direct chemical shift change in the presence of increasing NaCl concentration (Fig. 4a, b).

The salt sensing residues (yellow) are distributed in helix 2 and helix 4, on the surface perpendicular to the YmoA-H-NS interface (Fig. 4c, d). All salt sensing residues are highly conserved, except for The-61 and His-65 in YmoA, corresponding to Ser-66, and Phe-70 in Hha. Noticeably, helix 4 is involved in sensing both temperature and ionic strength, so are Arg-21, and His-65 (Fig. 4e, g).

The complete collection of charged residues were marked in cyan on the surface (Fig. 4f) to compare them to the distribution of salt-sensing residues (Fig. 4d). There are significantly more charged residues that potentially have the ability to bind to ions than salt sensing residues, particularly in helix one. This indicates that salt sensing is an event with high specificity, not merely due to the charged properties.

To see the dynamic effect of YmoA in response to different salt concentrations, we performed NMR relaxation of YmoA in the presence of low

salt (16 mM NaCl) and high salt (700 mM NaCl) (Fig. S7). We observed that most of the protein exhibits little or no change in its dynamics as measured from NMR transverse relaxation time ($R_2$) except on the C-terminus and precisely on Lys-67.

### Residues Arg-21 and His-65 are crucial for YmoA's function in sensing and responding to environmental changes

To validate the role of the amino acids identified as environmental sensors, we mutated two residues (Arg-21 and His-65) that showed significant chemical shift changes both upon NaCl binding and temperature increase. We then performed $R_2$ relaxation measurements in the presence or absence of high salt concentration (700 mM NaCl). These NMR experiments were conducted under the premise that the *ymoA* R21A and *ymoA* H65A mutants showed no significant differences in purification and stability (Fig. S8). Changes in $R_2$ relaxation time is often considered as a result of exchange in µs-ms timescale and it is often associated with conformational changes related to protein function (Palmer [33]). In this scenario, interaction with salt results in exchange and adaption of residues in the vicinity of this interaction. Here, we show that residue Lys-67 which itself does not interact directly with salt, experiences a significant broadening of $R_2$ when His-65 salt sensing residue is mutated, shifting the exchange from fast ns/ps to µs/ms (above detection limit) in high salt conditions. (Fig. 5a, b and S7). The R21A mutation did not affect the relaxation time of any of the residues compared to the wildtype.

We explored how mutations in specific residues (Arg-21, His-65, Lys-67) affect bacterial phenotype by trans-complementing the $\Delta ymoA$ mutant

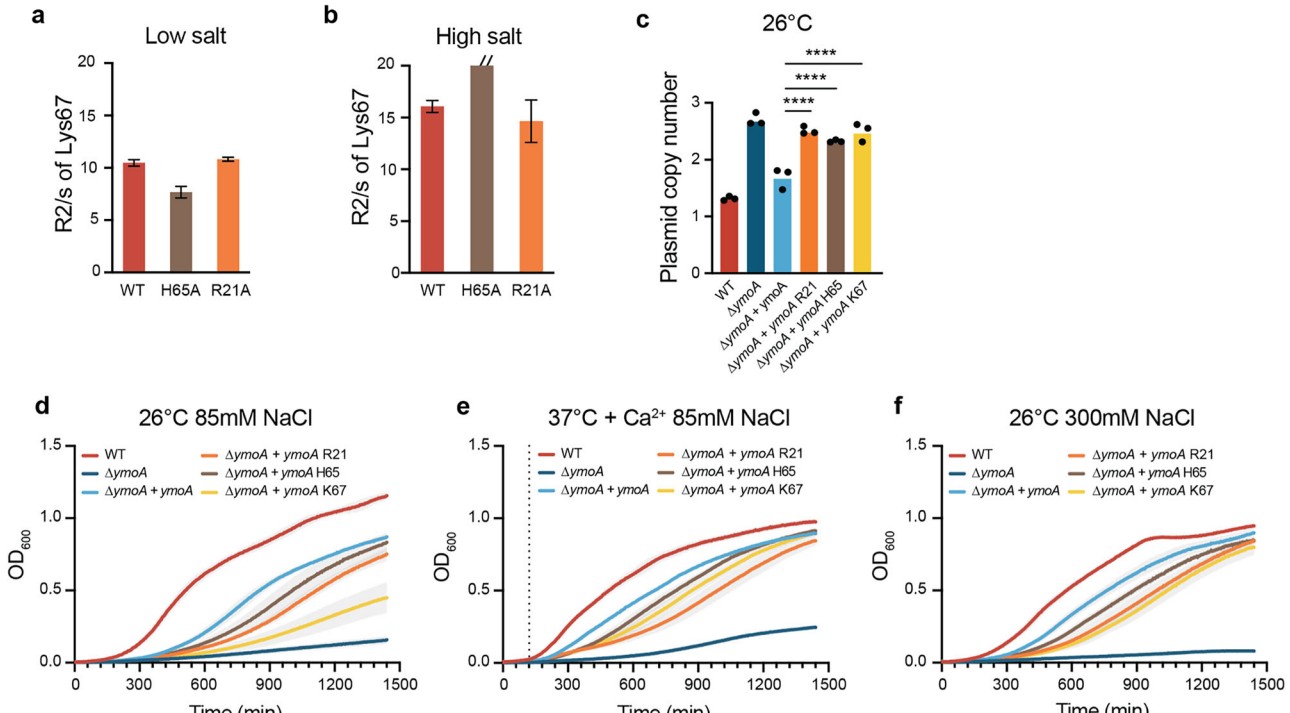

**Fig. 5 | YmoA C-terminus is essential for bacterial response to environmental changes.** NMR R2 relaxation measurement of the Lys67 residue in **a** low salt 75 mM or **b** high salt 500 mM concentration. **c–f** Strains were grown at (i) 26 °C for 5 h or (ii) 2 h at 26 °C and then shifted to 37 °C with or without $Ca^{2+}$ for 3 h. **c** PCN was determined using ddPCR. The data represent means ± SD ($n = 3$) ($^{**}P < 0.01$; $^{****}P < 0,0001$ using one-way Anova post hoc: Dunett's test) (**d–f**). Growth was determined by measuring the $OD_{600}$. The dotted line represented the time at which the temperature was switched from 26 °C to 37 °C. The data represent means ($n = 3$).

with *ymoA* R21A, *ymoA* H65A, and *ymoA* K67A. The PCN at 26 °C increased significantly between the Δ*ymoA* mutant trans-complemented with native *ymoA* and its mutated forms (Fig. 5c). At 26 °C, all three mutations resulted in a significantly reduced growth rate compared to the Δ*ymoA* mutant trans-complemented with the native *ymoA*. The Δ*ymoA* mutant trans-complemented with *ymoA* K67A showed growth rates nearly matching the uncomplemented Δ*ymoA* mutant (ns difference) (Fig. 5d). Under T3SS suppressive conditions at 37 °C, while the mutant complemented with the mutated residues still grew better than the uncomplemented Δ*ymoA* mutant, no significant growth differences were noted among them (Fig. 5e). A similar trend was observed with increased NaCl concentration (85 mM to 300 mM), except between the His-65 and Lys-67 mutations ($p = 0,0185$) (Fig. 5f, Supplementary Table 1). These mutations, affecting dual-sensing amino acids, resulted in significant decrease in the indicator's flexibility, slower growth, and increased PCN, confirming the role of YmoA, and specific amino acids, as environmental sensor and signal transmitter.

## Discussion

The ability to rapidly adapt to different environments is crucial for successful bacterial infections. In *Yersinia*, a key virulence factor, the T3SS, is tightly regulated at different levels to enable rapid activation upon host-cell contact[34]. Central to this regulation is YmoA, a small histone-like protein (Cornelis et al.[8]), which functions in an heterocomplex with H-NS[22]. H-NS is a DNA-binding protein known for binding AT-rich, curved DNA and important for gene regulation in Enterobacteriaceae. Together with H-NS, YmoA regulates numerous genes in *Yersinia*. Our study revealed that YmoA also critically regulates copy number of the virulence plasmid, facilitating rapid gene dosage adjustments for plasmid-encoded T3SS genes. Moreover, we elucidated at the molecular level the mechanism by which YmoA acts as an environmental sensor and further impacts the regulation of gene expression and virulence output. We propose that the regulatory cascade, initiated by environmental shift, followed by YmoA sensing, structural

changes in YmoA, altered binding affinity changes between YmoA and H-NS, and conformational changes in H-NS, is critical for downstream gene expression regulation. In this study, we primarily focused on how YmoA undergoes structural fluctuations and conformational changes in response to environmental cues such as temperature and osmolarity changes, and how these changes impact bacterial fitness and virulence functions.

YmoA belongs to a protein class found across various Enterobacteriaceae, including Hha in *E. coli*, *Salmonella* and *Klebsiella*. These proteins are predominantly conserved in their amino-acid sequences and overall structures, characterized as alpha-helical proteins with three helices forming a helix bundle, and the fourth helix positioned perpendicularly.

Our study has identified notable variations in the C-terminus region between YmoA and Hha from other species. Specifically, the fourth helix of YmoA exhibits an approximate 15-degree greater pivot compared to Hha from *E. coli* and *Salmonella*, when the main domains of three proteins are superimposed[26,27,30]. Additionally, we found that helix four undergoes conformational changes in response to both temperature and osmotic stress, underlining the importance of this helix for YmoA in *Yersinia*. The C-terminus is known to be important for the interaction with H-NS and a change of conformation in this region might explain its importance in YmoA function[35]. Our NMR analysis has successfully identified several specific residues (Arg-21, His-65, Lys-67) that are capable of sensing environmental changes. Mutations in these residues significantly impacted the cell growth and PCN in the bacteria, confirming their roles as environmental sensors (Fig. 5).

Furthermore, in Hha from *E. coli*, *Salmonella* and *Klebsiella* interact with H-NS via three amino acids, Glu-25, Glu-41, and Asp-48, while in YmoA, Glu-41 is substituted by Ala-36 and the other two amino acids remain conserved. This structural deviation, particularly the presence of a non-charged residue at one of the binding sites, could account for the distinct modulating mechanism observed in YmoA from *Yersinia*.

The successful pathogenicity of *Yersinia* depends largely on genes encoded on the pYV virulence plasmid, particularly the T3SS[17,18]. These

genes are subjected to both transcriptional and post-transcriptional regulation. Central to this process is LcrF, a transcriptional activator, which is in turn modulated by YmoA in a thermo-sensitive manner[19]. Additionally, recent research has unveiled a novel gene expression regulation mechanism in *Yersinia*, which involves rapid modulation of plasmid-encoded gene expression through changes in PCN[28,29]. Here, our study demonstrates that YmoA is pivotal in regulating temperature-mediated increases in PCN. This aligns with our observations that a temperature rise induces conformational changes in YmoA, altering its interaction with H-NS (Fig. 3). Therefore, YmoA is essential not only in regulating key genes, but also in controlling virulence plasmid replication in response to temperature fluctuations, adding an additional layer of gene regulation in *Yersinia*.

YmoA regulatory role extends beyond the T3SS, influencing a wide array of other virulence genes in response to diverse environmental stresses, including oxidative stress, starvation, and pH changes. For instance, YmoA was found to regulate the carbon storage regulator (Csr) system, which is important for the initiation of intestinal colonization and subsequent dissemination into deeper tissues[31]. In our study, we confirmed that the deletion of *ymoA* led to the overexpression of all plasmid-encoded genes. Interestingly, genes associated with the environmental life-style of the bacteria, such as those involved in flagellar motility, were down-regulated in the Δ*ymoA* mutant. These findings suggest that YmoA differentially regulates the expressions of genes involved in infection and housekeeping functions, adapting to various environmental cues.

The prevailing model suggests that YmoA undergoes thermo-regulation at the post-translational level, mediated by the ClpXP and Lon proteases. At 26 °C, YmoA remains stable, but it is degraded at 37 °C, which facilitates the expression of LcrF and subsequently the activation of the virulence genes at elevated temperatures[20]. The exact mechanism by which YmoA becomes a target for these proteases at 37 °C while remaining unaffected at 26 °C is not yet fully understood. One hypothesis is that a temperature-induced conformational change in YmoA at 37 °C could render it more susceptible to degradation.

In both *E. coli* and *Salmonella*, as well as in *Yersinia*, the genes *hha* and *ymoA* are part of an operon together with a second gene, named *tomb* and *ymoB*, respectively. Previous studies have indicated that Hha in *E. coli* and *Salmonella* exhibits conditional toxicity, while overexpression of *tomB* can mitigate Hha's toxicity. Hence, it has been proposed that Hha and TomB constitute a toxin-antitoxin (TA) system, with Hha acting as the toxin and TomB the antitoxin[12,13,36]. YmoB from *Yersinia* has demonstrated an antitoxin effect on Hha from *E. coli*, suggesting the presence of a similar TA system[13]. However, our findings diverge from this theory; overexpression of YmoA in *Yersinia* does not result in toxicity. Instead, YmoA expression is essential for optimal growth of *Yersinia* at 26 °C (Fig. 2g). These observations highlight the possibility that, despite belonging to a highly conserved protein family, YmoA and Hha may differ significantly in their functions.

Unlike TomB in *E. coli*, YmoB in *Yersinia* does not function as an antitoxin. A Δ*ymoB* mutant displayed no significant differences in either PCN or the expression of Yop effector proteins compared to wildtype under various conditions (Fig. 2c, d). Interestingly, trans-complementation of p*ymoB* resulted in an increase in the PCN in both the Δ*ymoA* mutant and the Δ*ymoA-ymoB* mutant. Additionally, a Δ*ymoB* mutant displayed reduced YmoA expression (Fig. S6). Together, these results suggest that YmoB may have a secondary, yet unclear role in the regulation of YmoA in *Yersinia*.

Collectively, our data suggest that YmoA response to environmental factors involves local adjustments of different residues. These adjustments result in the fine-tuned affinity shift between H-NS and YmoA, modulating gene expression and ultimately impacting the viability of the cells.

## Materials and methods
### Bacterial strains, plasmids, and growth conditions
Routine growth of *Yersinia pseudotuberculosis* (YPIII, Xen4 (pCD1 with Tn1000::Tn5 luxCDABE, Kan[R37]) strain was carried out at 26 °C in Lennox

agar or broth (10 g/L Tryptone, 5 g/L Yeast extract, 5 g/L NaCl) supplemented with appropriate antibiotics (kanamycin (Kan) 50 μg.ml$^{-1}$; chloramphenicol (Chl) 30 μg.ml$^{-1}$; ampicillin (Amp) 100 μg.ml$^{-1}$). For the assays, the strains were grown in one of the three following conditions (i) 26 °C; or grown for 2 h at 26 °C and then shifted to 37 °C (ii) with calcium; or (iii) without calcium. T3SS inductive conditions (low calcium) were achieved by adding 5 mM EGTA and 20 mM MgCl$_2$, and T3SS repressive conditions (high calcium) by adding 2.5 mM CaCl$_2$ and 20 mM MgCl$_2$. Bacteria carrying pBAD derivatives were grown in medium containing 0.1% arabinose for induction and the appropriate antibiotic selection.

### Construction of bacterial strains
**Mutant construction.** In frame deletions of *ymoB*, *ymoA*, and *ymoB-ymoA* in YPIII (Xen4) were constructed by homologous recombination using pDM4 as previously described (Olsson, Paulsson, and Nordström[38]). Briefly, three 1000 bp DNA fragments encoding 500 bp upstream and downstream sequences including the first 10 and last 10 codons of *ymoB*, *ymoA*, or a *ymoB-ymoA* fusion respectively, were constructed by overlapping PCR (for primers see Supplementary Table 5). The constructs were cloned into pDM4 using *sphI* and electroporated into *E. coli* S17-1 λ-pir. The final suicide plasmid was conjugated into YPIII (Xen4) and recombinant clones with the suicide plasmid integrated in the chromosome were recovered on LA plates containing 50 μg/ml Kan and 34 μg/ml Chl. In frame deletion mutants were recovered after SacB counter-selection on LA plates with 10% sucrose and 50 μg/ml Kan. The final mutants were verified by sequencing.

**Trans-complementation of the mutants.** The different genes of interest were amplified by PCR. Both the amplified genes and the pBAD18 vector were digested using EcoRI/XbaI Fast Digest enzymes. The inserts were cloned into the vector by ligation using T4 DNA ligase. Recombinant plasmids were transformed into *E. coli* DH5α electrocompetent cells and later into the YpIII strain of interest. Clones were confirmed by PCR using vector-specific primers and subsequently sent for Sanger sequencing. The point mutation of YmoA_R21A, H65A, and K67A were introduced into the native YmoA in pBAD18 using the Quick Change II kit (Agilent) by following the manufacturer's instructions.

**Curing of virulence plasmid.** Curing of the Xen4 plasmid was done by selecting colonies able to grow at 37 °C with Ca$^{2+}$ depletion. Colonies that grew on the plate were re-streaked on both Ca$^{2+}$-depleted plates and Kanamycin-containing plates to confirm the loss of the plasmid.

**Protein purification.** The *ymoB-ymoA* operon of *Y. pseudotuberculosis* was amplified, introducing a C-terminal His$_6$-tag (for primers see Supplementary Table 5). The amplified operon was TOPO-ligated to pEXP5-CT vector (Invitrogen), and transformed to *E. coli* TOP10 (Invitrogen) chemically competent cells. The YmoB-YmoA-pEXP5-CT in cells was selected by 50 μg/ml carbenicillin (Cb), and extracted with QIAprep Spin Miniprep Kit (QIAGEN). The correctness of constructions was verified by sequencing (Eurofins). YmoB-YmoA-pEXP5-CT was chemically transformed to One Shot™ BL21 Star™ (DE3) (ThermoFisher Scientific) for expression. The colonies were cultured in 1 L Luria-Bertani (LB) broth medium with 50 μg/ml Cb at 37 °C with an agitation of 120 rpm. When the OD$_{600}$ of the culture reached 0.6–0.8, the overexpression was induced by the addition of 250 μM Isopropyl β-D-thiogalactoside (IPTG). The cell culture was incubated at 22 °C for 16 h. Protein purification followed the protocol described previously[30]. Briefly, the harvested cells were suspended with the lysis buffer A at a ratio of 5 ml/g cells (containing 50 mM sodium phosphate (pH 8.0) 300 mM NaCl, the Complete EDTA-free protease inhibitor tablet (1 tablet/50 ml buffer, Roche Molecular Biochemicals, Indianapolis, IN, 2 mM BME). The full suspension passed through the continuous cell disruptor at 30kpsi (TS series cabinet, Constant Systems Limited). The lysates were centrifuged at 25,000 × *g* for 40 min. The supernatant was applied to 1 ml Nickel resin

(Ni Sepharose 6 Fast Flow, GE Healthcare) preequilibrated with 20 ml buffer A and 20 mM imidazole, and incubated at 4 °C for 1 h. The resin was washed with 10 ml buffer A plus 20 mM imidazole, and 5 ml buffer A plus 50 mM imidazole. The bound protein was eluted with 3 ml buffer A plus 250 mM imidazole. The eluate was loaded to PD-10 column pre-equilibrated with the final buffer (50 mM Potassium phosphate (pH 6.0), 150 mM NaCl, 5 mM dithiothreitol (DTT)), and eluted with the final buffer. Different samples were collected, and analyzed by SDS-PAGE for purity, western blotting and mass spectrometry for identification. The concentration of YmoA_his was determined by the absorbance at 280 nm with the extinction coefficient of 11460 (ProtParam). YmoA_his has a purity above 90%.

For YmoA_R21A and H65A, the same protocol was followed for purification. Mutations R21A and H65A were introduced with PCR amplifying the whole YmoB-YmoA-pEXP5-CT using the Quick Change II kit (Agilent) and following user instructions (for primers see Supplementary Table 6).

**Isotope incorporation.** In order to incorporate $^{13}C$ and $^{15}N$ to YmoA for NMR spectroscopy, the cells with YmoB-YmoA-pEXP5-CT in LB medium were grown to mid-log phase ($OD_{600}$ 0.5), and centrifuged at $4000 \times g$ for 10 min. The cell paste was washed twice with M9 minimal salt medium plus 1 g/L $^{13}C$ glucose, and 1 g/L $NH_4Cl$, and resuspended with the same medium. The cells were grown at 37 °C with an agitation of 120 rpm, and the over expression was induced with 250 μM IPTG, when the $OD_{600}$ reached 0.8–0.9. The protein purification followed the protocol described above. The purified YmoA_his was concentrated with a Vivaspin Turbo concentrator with a cutoff of 5 kDa. The $^{13}C/^{15}N$-labeled YmoA was concentrated to roughly 1 mM, and ready for NMR spectroscopy.

**Circular Dichroism.** Far-UV CD spectra were collected in a spectro-photometer (J-1500 CD spectrometer, JASCO) with a quartz cuvette of path length 5 mm. The signal was presented in mean residue ellipticity (MRE) unit of degree*$cm^2$*$dmol^{-1}$. The optimum YmoA_his concentration for NMR was determined as 16 μM with Initial tests. The CD spectra were collected under various conditions specifically in NaCl concentration gradient at a range of 37.5 and 500 mM, and in temperatures from 18 to 75 °C, to assess the stability of YmoA_his in regard to secondary structure.

**Thermal shift assay.** TSA was conducted in (CFX Cpnnect™ real-time system, BIO-RAD) to analyze the denaturation of YmoA_his in response to temperature. YmoA_his was mixed with 1X SYPRO orange dye (BIO-RAD) at the final concentration of 20 μM. The assay was run in a format of 96-well plate with the temperature gradient from 15 to 95 °C. The first derivative of fluorescence emission was plotted as a function of temperature to identify the melting temperature at the inversion point of the curve.

**Bacterial fitness measurements**
To measure bacterial growth, overnight cultures were diluted 1:500 into the chosen culture media with required selection and induction. The $OD_{600}$ was measured every 4 min in a total time course of 16 h or 24 h under continuous shaking in a Bioscreen C MBR (Oy Growth Curves Ab Ltd). Growth rate values were calculated using the BAT2.1 online tool (Thulin, 2018). Data were analyzed using Prism 9.

**Plasmid copy number determination by digital droplet PCR**
Overnight cultures were diluted 1:50 and grown either 5 h at 26 °C or 2 h at 26 °C and then shifted to 37 °C for 3 h with or without calcium. After incubation, samples were taken, spun down and the supernatant was removed. Whole genome DNA was extracted using GeneJET Genomic DNA purification Kit (ThermoScientific) following the recommended protocol. The concentration of the eluted DNA was determined using a

Qubit 2.0 fluorometer (Thermo Scientific). ddPCR was carried out as described in ref. 28 using one couple of primers for the plasmid and another one for the chromosome. Data were analyzed using Prism 9.

**Yop expression and Yop secretion profile**
*Y. pseudotuberculosis* YPIII Xen4 overnight cultures were diluted either 1:20 (wild-type and Δ*ymoB*) or 1:10 (Δ*ymoA* and Δ*ymoBymoA*) in fresh media, with selection and induction if required, in order to obtain a similar OD600 after incubation. The cells were grown 2 h at 26 °C, then, they were either left at 26°C or shifted to 37°C for 1 more hours. For the YmoA degradation assay, cells were sampled at 1, 2 and 3 h after being shifted to T3SS-inductive conditions (37 °C without $Ca^{2+}$). At indicated time points, $OD_{600}$ was measured and samples equivalent to $10^9$ cells were taken. Both the pellet and the supernatant were mixed with 4xLaemmli SDS sample buffer (BIO-RAD), and 2-Mercaptoethanol 1:10 and heated up to 98 °C for 5 min. Protein samples were subsequently separated on a 4-20% gradient Mini Protean TGX Precast Gel (BIO-RAD). Whole cell lysate samples were all loaded with the same volume, whereas the volume of the supernatant samples was adjusted according to their $OD_{600}$ at the respective time point. The gel was furthermore analyzed using silver-staining or western blotting. For the silver stain, a Pierce Silver Stain Kit (Thermo Scientific) was used according to the standard protocol. The samples were transferred onto a Trans-Blot Turbo Mini 0.2 μm PVDF Transfer Pack (BIO-RAD) and probed using Anti-YmoA or anti-Yops/YopE polyclonal antibodies (Agrisera). Proteins of interest were detected using Chemiluminescence by adding Amersham Enhanced chemiluminescence (ECL) detection reagent (GE healthcare). For quantification of Yops-secretion, all anti-Yops bands of each sample were analyzed using the density tool of the software.

**Isolation of bacterial RNA**
Three independent cultures of *Y. pseudotuberculosis* wildtype strain along with single deletional mutant of *ymoA* and *ymoB* were grown at 26 °C overnight. Before RNA extraction, overnight cultures were diluted 1:25 into fresh media and cultured for 5 h at 26 °C. Total RNA (three biological replicates per strain) was extracted using a Trizol RNA isolation kit according to the manufacturer's protocol. RNA concentration was measured using Qubit® RNA Assay Kit in Qubit® 2.0 Flurometer (Life Technologies, CA, USA).

RNA was sent to Novogene Bioinformatics Institute, there, RNA degradation and contamination was monitored on 1% agarose gels. RNA purity was checked using the NanoPhotometer® spectrophotometer (IMPLEN, CA, USA). RNA integrity and quantitation was assessed using the RNA Nano 6000 Assay Kit of the Bioanalyzer 2100 system (Agilent Technologies, CA, USA).

**Library preparation for transcriptome analysis, clustering, and sequencing**
A total amount of 1 μg RNA per sample was used as input material for the RNA sample preparations. Sequencing libraries were generated at the Novogene Bioinformatics Institute using NEBNext® Ultra™ RNA Library Prep Kit for Illumina® (NEB, USA) following manufacturer's recommendations and index codes were added to attribute sequences to each sample. Briefly, mRNA was purified from total RNA using poly-T oligo-attached magnetic beads. Fragmentation was carried out using divalent cations under elevated temperature in NEBNext First Strand Synthesis Reaction Buffer(5X). First strand cDNA was synthesized using random hexamer primer and M-MuLV Reverse Transcriptase (RNase H). Second strand cDNA synthesis was subsequently performed using DNA Polymerase I and RNase H. Remaining overhangs were converted into blunt ends via exonuclease/polymerase activities. After adenylation of 3' ends of DNA fragments, NEBNext Adapter with hairpin loop structure were ligated to prepare for hybridization. In order to select cDNA fragments of preferentially 150 ~ 200 bp in length, the library fragments were purified with AMPure XP system (Beckman Coulter, Beverly, USA). Then 3 μl USER Enzyme (NEB, USA) was used with size-selected, adapter-ligated cDNA at 37 °C for

15 min followed by 5 min at 95 °C before PCR. Then PCR was performed with Phusion High-Fidelity DNA polymerase, Universal PCR primers, and Index (X) Primer. At last, PCR products were purified (AMPure XP system) and library quality was assessed on the Agilent Bioanalyzer 2100 system.

The clustering of the index-coded samples was performed on a cBot Cluster Generation System using TruSeq PE Cluster Kit v3-cBot-HS (Illumina) according to the manufacturer's instructions. After cluster generation, the library preparations were sequenced on an Illumina Hiseq 2000 platform and 100 bp paired-end reads were generated.

### RNAseq data analysis

Data analysis was done by Novogene Bioinformatics Institute. Raw data (raw reads) of FASTQ format were firstly processed through fastp. In this step, clean data (clean reads) were obtained by trimming reads containing adapter and removing poly-N sequences and reads with low quality from raw data. At the same time, Q20, Q30, and GC content of the clean data were calculated. All the downstream analyses were based on the clean data with high quality.

Reference genome and gene model annotation files were downloaded from genome website directly. Both building index of reference genome and aligning clean reads to reference genome were used Bowtie2. (Langmead, B. and S.L. Salzberg, 2012)

FeatureCounts was used to count the reads numbers mapped to each gene. And then FPKM of each gene was calculated based on the length of the gene and reads count mapped to this gene. FPKM, expected number of Fragments Per Kilobase of transcript sequence per Millions base pairs sequenced, considers the effect of sequencing depth and gene length for the reads count at the same time, and is currently the most commonly used method for estimating gene expression levels.

(For DESeq2 with biological replicates) Differential expression analysis of two conditions/groups (two biological replicates per condition) was performed using the DESeq2 Rpackage. DESeq2 provide statistical routines for determining differential expression in digital gene expression data using a model based on the negative binomial distribution. The resulting $P$ values were adjusted using the Benjamini and Hochberg's approach for controlling the false discovery rate. Genes with an adjusted $P < 0.05$ found by DESeq were assigned as differentially expressed.

(For edgeR without biological replicates) Prior to differential gene expression analysis, for each sequenced library, the read counts were adjusted by Trimmed Mean of Mvalues (TMM) through one scaling normalized factor. Differential expression analysis of two conditions was performed using the edgeR R package. The $P$ values were adjusted using the Benjamini and Hochberg methods. Corrected $p$ value of 0.005 and |log2(Fold Change)| of 1 were set as the threshold for significantly differential expression.

Gene Ontology (GO) enrichment analysis of differentially expressed genes was implemented by the clusterProfiler R package, in which gene length bias was corrected. GO terms with corrected $P$ value less than 0.05 were considered significantly enriched by differential expressed genes.

KEGG is a database resource for understanding high-level functions and utilities of the biological system, such as the cell, the organism, and the ecosystem, from molecular level information, especially large-scale molecular datasets generated by genome sequencing and other high-through put experimental technologies (http://www.genome.jp/kegg/). We used clusterProfiler R package to test the statistical enrichment of differential expression genes in KEGG pathways.

### NMR spectroscopy temperature dependent chemical shift analysis

All NMR spectroscopy experiments were done on a Bruker NeoAdvance 600 MHz spectrometer equipped with a cryogenic TCI probe (CRPHe TR-1H &19F/13C/15N 5mm-EZ). YmoA protein concentration used range between 0.5mM and 0.7 mM. All NMR experiments were done at 298 K except for the temperature dependent chemical shift monitoring experiment which was done between 288 - 320 K. Initial an HNCACB and HNcoCACB experiment was done to confirm the amino acid backbone assignment. For the temperature dependent experiment, YmoA was dissolved in 50 mM Sodium phosphate pH 6.5, 150 mM NaCl. The sample was supplemented with 10% D2O and 0.1% Sodium azide. Salt dependent experiments were done with protein dissolved in 50 mM Sodium phosphate pH 6.5 150 mM NaCl as the starting protein. Increasing concentration of NaCl was titrated to the sample to a final concentric of 700 mM. Additionally, NMR relaxation was performed for YmoA in the presence of 700 mM NaCl, 16 mM NaCl and as well as 150 mM NaCl. Relaxation experiments were performed on respective YmoA mutants at low salt (16 mM NaCl) and at high salt (700 mM NaCl). For these experiments, the relaxation delay was set to 4 s and 6 randomized relaxation times were used for the respective experiments. For all the salt experiments the protein solution was buffered with 50 mM sodium phosphate pH 6.5. All NMR experiments were processed with Topspin version 4 series and analyzed using ccpNMR analysis software. All relaxation data was analyzed with DynamicCenter version 2.8.01. For estimation of the amide hydrogen bonding, amide chemical shifts were compared to those of random coil and plotted as a function of amino acid sequence as in ref. 39.

### Statistics and reproducibility

All experiments were carried out independently and repeated at least in triplicate to ensure reproducibility. Results are expressed as mean ± standard deviation (SD) derived from multiple independent experiments. All statistical analysis was done using Prism 10, and the criteria for significance and statistical tests employed are detailed within the respective sections.

### Data availability

The RNAseq data have been deposited in NCBI with the GEO accession number GSE280333 and BioProject number PRJNA1177866. All uncropped Western blots, gels and RNA seq analysis are provided in the Supplementary data.

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

## Acknowledgements
This work was supported by grants from the Swedish Society for Medical Research (S18-0174), the Swedish Research Council (2018-02376, 2022-00741), and the Carl Trygger Foundation (CTS20:458) to H.W. We sincerely thank Robin Lissner for his contributions during his Master's project in the H.W. lab and for his outstanding documentation skills.

## Author contributions
T.H., T.E., and H.W. designed the study. H.T., L.L., T.E., and C.C. performed the experiments and data analyses. All authors contributed to writing of the manuscript.

## Funding

## Competing interests
The authors declare no competing interests.
