## [Transparent Peer Review file · Communications Biology]

YmoA functions as a molecular stress sensor in Yersinia

Corresponding Author: Professor Helen Wang

Version 0:

Reviewer comments:

Reviewer #1

(Remarks to the Author)

Hha family of proteins are implicated in numerous stress related responses ranging from temperature, and pH to osmotic stress. Together with H-NS, they regulate numerous stress-related genes thus modulating the survival and fitness of numerous bacterial including Yersinia. In this work, Hechard et al employ the protein YmoA from Yersinia to explore the extent the conformational changes observable in YmoA on changing salt/temperature. This is followed by RNA-seq analysis and plasmid copy number determination apart from conventional mol.bio. techniques to establish the stress regulatory features of YmoA.

The study is systematic, and the manuscript is also well-written. However, several aspects need to be addressed before publication.

1) It would be informative to see a thermal melt (from far-UV CD at 222 nm) to understand the signal changes in the low temperature regime at the two different salt conditions studied. It is difficult to observe any changes in the far-UV CD spectra provided in the supporting information. Please show the thermal melt in the main text at the different salt concentrations.

2) It is surprising that Coredeiro et al (JBC, 2015) report large changes in chemical shifts and relaxation dynamics in their work, while no such changes in dynamics are observed in YmoA from Yersinia, despite having a near-identical sequence. Any reason for this?

3) Do then the chemical shifts reported in Figure 4c arise from ion binding or from changes in structure? It is critical to address this question, as they are two different potential mechanisms.

4) Figure 4b: It is hard to decide if the chemical shift differences are significant given the minor changes. Report error bars for the chemical shifts (from some intrinsic uncertainty in measurements) without which it is not possible to objectively judge. Same with Figure 3b.

5) Also, why is 0.05 a threshold for ΔCS and why not higher? Without a reference or a justification it is again quite subjective.

6) One aspect that is gaining traction is the role of such Hha-family proteins in regulating the oligomerization extents of H-NS which is the repressor (see DOI: 10.1093/nar/gkae090). These could be discussed in the concluding sections, as it is possible that it is not just the structural change or dynamics in YmoA, but the binding to H-NS which could be effectively determine the extent of oligomerization and hence regulate transcription/virulence.

7) Do the authors observe any changes in H-NS expression profiles in the conditions/mutants studied?

Minor comments:

1) A single concentration unit is preferred. CD is reported to be studied at 0.15 mg/ml while NMR at 0.5-0.7 mM. It would be preferable that mM or microM is used. What is the pH of the buffer solution used for CD?

2) The secondary structure annotation should also be provided for 4b, similar to Figure 1a.

Reviewer #2

(Remarks to the Author)

In their manuscript entitled „YmoA functions as a molecular stress sensor in Yersinia” submitted to Communications Biology (#: COMMSBIO-24-3434), Héchard and coworkers investigate the molecular mechanisms by which temperature is transduced via YmoA in gene regulation of Yersinia.

Although the authors address an interesting topic and provide new data, the manuscript has several severe drawbacks that need to be addressed carefully. In particular, results and methods are insufficiently described, data are overinterpreted, and the manuscript is poorly written in large parts. In general, the manuscript does not pass scientific standards.

Major points:

1. First results section 2.1 and Fig. 1: Large parts, in particular “In terms of amino acids..” to the end of this chapter (“...in the C-terminus”) are literature data rather than own results, please rewrite accordingly. Then, the authors mention and compare the amino acid sequences of YmoA from five bacteria only. However, this is not state of the art with respect to bioinformatic tools. Please provide a much more extensive analysis of all or at least most relevant genera belonging to Enterobacteriaceae with respect to a YmoA sequence comparison, and provide a phylogenetic tree thereof.
2. Results section 2.2 must be rewritten and should show much more details. The authors should first describe what they were doing here. Obviously, they intend to delineate the transcriptome of a set of strains under certain conditions, this must be mentioned here. However, they cite Böhme et al. in the second sentence, so this is not new data?
3. Figure S3: The head is misleading, should read “...in deletion mutants of ymoA and ymoB.”
4. The data presentation is not state of the art. An overview is required that spells the gene functions grouped according to operons and their function. Much more details about the transcriptome analysis must be mentioned.
5. The authors probably use a *Y. pseudotuberculosis* strain for their experiments, but the species is not mentioned at all in the results!
6. Page 5 of the manuscript: Should read...were both slightly expressed...; “YmoA contributes to the regulation of T3SS expression at 26°C”.
7. Result section 2.4: Method is missing.
8. Result section 2.5: please improve introduction and rewrite first sentence, why do the authors address this point?
9. Result section 2.8: Method to introduce point mutations is missing.
10. YmoA not only plays a role in gene regulation at body, but also at environmental temperature, please cite Starke&Fuchs Mol Micro 2014 (doi: 10.1111/mmi.12554)

Minor points:

1. Please indicate line numbers.
2. In a potential revision process, please provide a marked-up manuscript.
3. Abstract should read “the molecular mechanisms transducing environmental signals...”
4. Results section 2.2: Please provide a reference for the first sentence
5. Fig. S3 should be shown as a full Figure.
6. Gene names must be written in italics.
7. Order supplementary figures according to their appearance in the text.
8. Page 5 of the PDF and elsewhere: “YmoA is known...”, “YmoA is essential...”: please provide references.

Reviewer #3

(Remarks to the Author)

Héchard et al. conducted a comprehensive structure-function analysis of YmoA, a Yersinia transcription factor that responds to various environmental changes. Using NMR spectroscopy, they identified key residues in the C-terminal region of YmoA that respond to temperature and/or salt fluctuations. The study is complemented by RNA-sequencing experiments mapping the YmoA regulon and by phenotypic analyses of an ymoA deletion strain and complementation experiments with point-mutated YmoA variants.

Questions and comments:

1. Providing page and/or line numbers would facilitate reviewing.
2. The manuscript reports that YmoA plays a regulatory role in plasmid copy number control and in expression and secretion of Yops. What are the underlying molecular mechanisms? Are copy number control and Yop control distinct mechanisms? Are they direct or indirect effects exerted through YmoA-regulated factors?
3. A figure summarizing the conclusions would help.
4. Does salt influence the plasmid copy number or Yop secretion?
5. Does salt affect degradation of YmoA?
6. Have the RNAseq results been deposited in a public database, like Gene Expression Omnibus (GEO)?
7. Fig. 2a: It would be interesting to have some selected transcripts, e.g. lcrF, indicated in the volcano plot.
8. Fig. 2i: Can the salt defect be complemented?

9. References require careful editing, e.g. 7 times (Zhou et al. 1992) towards the end of Material and Methods. Reference "Wang, He" should be "Wang, Helen". Once it is "Cornelis, G.R.", then it is "Cornelis, Guy R.". Some journals are abbreviated, others not, and so on and so forth.

Version 1:

Reviewer comments:

Reviewer #1

(Remarks to the Author)

The authors have addressed my concerns and questions. I recommend publication after the minor revision below.

Just one note of caution on the secondary structure content at different ionic strength values: while the spectra indicate that the secondary structure contents are near-identical, they do not provide information on whether stability of the protein is altered. If altered, then it would mean that the interaction patterns are different across the different ionic strength conditions. The authors could directly check for it, by monitoring the signal at 222 nm (far-UV CD) as a function of temperature, instead of a thermal shift assay which is a more complex way to measure stability changes.

Reviewer #2

(Remarks to the Author)

I appreciate that the authors addressed the points made by this reviewer. However, probably due to a misunderstanding, one major point on transcriptome data still requires a major revision.

Presentation of the transcriptome data is not state of the art and should be revised. First, with few exceptions, the data mentioned in the results section 2.2 is too short, and the figure 2 is too descriptive. The authors should give more details about genes beside T3SS and virulence factor. To this end, a more thorough analysis and much better presentation of the data of figure S2b is required that shows the gene functions grouped according to operons and their function. Much more details about the transcriptome analysis must be mentioned. The whole data set should be provided as supplementary material. What is the significance level, how often were the experiments repeated, why do the authors show only 20 genes most strongly up- or downregulated? Genes involved in virulence or plasmid-encoded should be group. Genes of unknown function should be manually reannotated to provide more functional information.

Reviewer #3

(Remarks to the Author)

I am satisfied with the revision

Response to reviewers' comments

Reviewer #1

1) It would be informative to see a thermal melt (from far-UV CD at 222 nm) to understand the signal changes in the low temperature regime at the two different salt conditions studied. It is difficult to observe any changes in the far-UV CD spectra provided in the supporting information. Please show the thermal melt in the main text at the different salt concentrations.

To address the reviewer's comments, we have performed thermal shift assay on YmoA with three different salt concentrations, ranging from 37.5mM up to 500mM NaCl concentration. This is now presented in Fig S6b. In order to provide additional clarity for the readers, we have added the following statement to the result 2.5 section (Lines 213-215): "At the same temperature, YmoA possessed the same composition of the secondary structures, when it was tested with three different salt concentrations, ranging from 37.5 mM to 500 mM NaCl (**Fig. S6b**)."

2) It is surprising that Coredeiro et al (JBC, 2015) report large changes in chemical shifts and relaxation dynamics in their work, while no such changes in dynamics are observed in YmoA from Yersinia, despite having a near-identical sequence. Any reason for this?

We have observed several key differences between *E. coli* Hha and *Yersinia* YmoA. Firstly, there are notable sequence differences, particularly at the C-terminus, which is known to be crucial for the function of YmoA/Hha. Secondly, phenotypic differences are evident between *E. coli* and *Yersinia*, such as the effects on growth regulation when YmoA and Hha are overexpressed, and temperature-inducing virulence effector expression. Given these distinctions, it is not surprising that differences in chemical shift and relaxation dynamics were also observed in two different bacterial species.

3) Do then the chemical shifts reported in Figure 4c arise from ion binding or from changes in structure? It is critical to address this question, as they are two different potential mechanisms.

The chemical shifts observed arise from ion binding. The plot in the figure illustrates the difference in chemical shifts between low and high salt conditions, as indicated in the figure legend 4b. "b. A plot of the difference of the chemical shift change at low and high salt with respect to amino acid sequence. The chemical shift (CS) difference was computed according to the following equation ($CS = \text{SQRT}((\Delta N^2)/5 + \Delta H^2)$)."

4) Figure 4b: It is hard to decide if the chemical shift differences are significant given the minor changes. Report error bars for the chemical shifts (from some intrinsic uncertainty in measurements) without which it is not possible to objectively judge. Same with Figure 3b.

We did not initially estimate the error in the chemical shift differences, as shifts of 0.2-0.4 in both the 1H and 15N dimensions are generally considered significant. Typically, errors in these

measurements are estimated based on the accuracy of peak positions. In our current measurements, the estimated error is below 1%, derived from the noise in the spectrum. We have now included this error estimation in the plot, alongside a statement in the figure legend.

5) Also, why is 0.05 a threshold for deltaCS and why not higher? Without a reference or a justification it is again quite subjective.

The cutoff of 0.05 was chosen to be five times larger than the standard deviation (0.01) to ensure that only residues with significant changes were selected. A cutoff of 0.01 would include a few additional residues, primarily at the N-terminus and residues 18-35, but would not alter the overall conclusion, as these regions are already highlighted in Figure 4.

6) One aspect that is gaining traction is the role of such Hha-family proteins in regulating the oligomerization extents of H-NS which is the repressor (see DOI: 10.1093/nar/gkac090). These could be discussed in the concluding sections, as it is possible that it is not just the structural change or dynamics in YmoA, but the binding to H-NS which could be effectively determine the extent of oligomerization and hence regulate transcription/virulence.

We agree with the reviewer's perspective completely, that the structural changes of YmoA can affect its interaction with H-NS. It is highly likely that environmental cues are transmitted via YmoA to regulate gene expression and virulence output through interaction with H-NS. To address this comment, we revised the Discussion section and improved the clarity of this section (Line 322-330).

7) Do the authors observe any changes in H-NS expression profiles in the conditions/mutants studied?

It is a very interesting point; we have so far not observed any changes in H-NS expression with our RNA-seq data.

Minor comments:

1) A single concentration unit is preferred. CD is reported to be studied at 0.15 mg/ml while NMR at 0.5-0.7 mM. It would be preferable that mM or microM is used. What is the pH of the buffer solution used for CD?

Now this is changed from 0.15 mg/ml to 16 μ M (Line 491).

The pH of the buffer solution used for CD was 6.0, the same as the one used for NMR.

2) The secondary structure annotation should also be provided for 4b, similar to Figure 1a.

We have secondary structure annotated to the sequence in both NMR figures.

Reviewer #2

Major points:

- 1. First results section 2.1 and Fig. 1: Large parts, in particular “In terms of amino acids..” to the end of this chapter (“...in the C-terminus”) are literature data rather than own results, please rewrite accordingly. Then, the authors mention and compare the amino acid sequences of YmoA from five bacteria only. However, this is not state of the art with respect to bioinformatic tools. Please provide a much more extensive analysis of all or at least most relevant genera belonging to Enterobacteriaceae with respect to a YmoA sequence comparison, and provide a phylogenetic tree thereof.**

We appreciate reviewer 2's comments. In response, we have condensed and clarified this chapter, and have completely revised the result section 2.1. Additionally, it appears that reviewer 2 may have overlooked Figure S1, which presents the amino acids sequence alignment of YmoA/Hha across 30 most relevant species, along with a phylogenetic tree.

- 2. Results section 2.2 must be rewritten and should show much more details. The authors should first describe what they were doing here. Obviously, the intend to delineate the transcriptome of a set of strains under certain conditions, this must be mentioned here. However, the cite Böhme et al. in the second sentence, so this is not new data?**

In response, we have revised this part to increase clarity. Pls see Line 116-120.

- 3. Figure S3: The head is misleading, should read “...in deletion mutants of ymoA and ymoB.**

This was corrected accordingly.

- 4. The data presentation is not state of the art. An overview is required that spells the gene functions grouped according to operons and their function. Much more details about the transcriptome analysis must be mentioned.**

Please refer to Figure 2b and S3 for gene categorized by their respective functions. Please find the more details about the transcriptome analysis in the Materials and Methods section “RNAseq Data Analysis”.

- 5. The authors probably use a *Y. pseudotuberculosis* strain for their experiments, but the species is not mentioned at all in the results!**

We have added the species name in the results section accordingly.

- 6. Page 5 of the manuscript: Should read...were both slightly expressed...; “YmoA contributes to the regulation of T3SS expression at 26°C”.**

This was corrected accordingly (Line 155).

7. Result section 2.4: Method is missing.

The method was added accordingly (Line 521-523).

8. Result section 2.5: please improve introduction and rewrite first sentence, why do the authors address this point?

The first sentence in result section 2.5 has been revised accordingly. Ensuring correct folding and stability is essential to rule out any potential structural artifacts that could skew the interpretation of flexibility changes, thereby providing a reliable foundation for investigating environmental influences.

9. Result section 2.8: Method to introduce point mutations is missing.

"The point mutation of YmoA_R21A, H65A and K67A were introduced into the native YmoA in pBAD18 using the Quick Change II kit (Agilent) by following the manufacturer's instructions." The method is present in the section "**Construction of bacterial strains**" under "Trans-complementation of the mutants".

10. YmoA not only plays a role in gene regulation at body, but also at environmental temperature, please cite Starke&Fuchs Mol Micro 2014 (doi: 10.1111/mmi.12554)

This was added accordingly in Line 61-62.

Minor points:

1. Please indicate line numbers.

Done

2. In a potential revision process, please provide a marked-up manuscript.

Done

3. Abstract should read "the molecular mechanisms transducing environmental signals..."

Done

4. Results section 2.2: Please provide a reference for the first sentence.

Done

5. Fig. S3 should be shown as a full Figure.

Figure 2 was shown as a full figure

6. Gene names must be written in italics.

Done

7. Order supplementary figures according to their appearance in the text.

Changed accordingly

8. Page 5 of the PDF and elsewhere: "YmoA is known...", "YmoA is essential...": please provide references.

Changed accordingly

Reviewer #3

1. Providing page and/or line numbers would facilitate reviewing.

Page and line numbers are now added in the manuscript.

2. The manuscript reports that YmoA plays a regulatory role in plasmid copy number control and in expression and secretion of Yops. What are the underlying molecular mechanisms? Are copy number control and Yop control distinct mechanisms? Are they direct or indirect effects exerted through YmoA-regulated factors?

First, we would like to thank the reviewer for such insightful comments. In this study, we aim to investigate the molecular mechanism of YmoA as a central sensor and regulator for plasmid copy number and T3SS activation/Yop secretion, whether directly or indirectly. It is well-known that T3SS genes are regulated by the transcription activator LcrF, which is controlled by YmoA. While H-NS homodimers bind the *lcrF* promoter, full regulatory function requires YmoA. It is likely that YmoA influences LcrF by forming a complex with H-NS to modulate DNA supercoiling (Madrid, Nieto, and Juárez, 2002). In a $\Delta ymoA$ mutant, LcrF expression increases at 37°C, indicating that YmoA regulates T3SS genes in a temperature-dependent manner, independent of plasmid copy number. Our previous results suggest that YmoA regulates the plasmid copy number via *copA* (an anti-sense RNA controlling the translational rate of the plasmid initiator RepA), similar to the regulation seen with YopD (Engling et al., 2023), adding another layer of regulation to the Yersinia virulence network. However, the molecular details of this regulation remain unclear. The questions raised by the reviewer are central to our study. However, delineating the complex regulatory network involved in this process presents a considerable challenge, as they seem to involve several overlapping factors that affect each other.

3. A figure summarizing the conclusions would help.

We really appreciate the suggestion to include a new figure summarizing the conclusions of our paper. However, after careful consideration, we believe that attempting to condense the complexity of our findings and essential pathways involved into a single figure would oversimplify the key points and may not significantly enhance the clarity or impact of the manuscript. However, we have ensured that the conclusions are clearly articulated in the text.

4. Does salt influence the plasmid copy number or Yop secretion?

Increase in salt concentration from 75mM to 500mM increases the plasmid copy number very slightly in wildtype Yersinia strain, with no statistical significance.

5. Does salt affect degradation of YmoA?

This is a very interesting idea; however, we have not proceeded that, rather focusing on the degradation of YmoA in response to temperature changes.

6. Have the RNAseq results been deposited in a public database, like Gene Expression Omnibus (GEO)?

The RNAseq data have been deposited in NCBI with the GEO accession number GSE280333 and BioProject number PRJNA1177866. Pls refer to the Data availability section (Line 636-638).

7. Fig. 2a: It would be interesting to have some selected transcripts, e.g. lcrF, indicated in the volcano plot.

We have now indicated some selected transcripts related to the function of T3SS in the volcano plot.

8. Fig. 2i: Can the salt defect be complemented?

Different salt concentrations result in distinct phenotypes. The mutant *ymoA* was trans-complemented with the native *ymoA* and various point mutated versions at both the 85mM and 300mM NaCl concentrations.

9. References require careful editing, e.g. 7 times (Zhou et al. 1992) towards the end of Material and Methods. Reference “Wang, He” should be “Wang, Helen”. Once it is “Cornelis, G.R.”, then it is “Cornelis, Guy R.”. Some journals are abbreviated, others not, and so on and so forth.

This has been corrected accordingly.

Response to reviewers' comments

Reviewer #1

The authors have addressed my concerns and questions. I recommend publication after the minor revision below.

Just one note of caution on the secondary structure content at different ionic strength values: while the spectra indicate that the secondary structure contents are near-identical, they do not provide information on whether stability of the protein is altered. If altered, then it would mean that the interaction patterns are different across the different ionic strength conditions. The authors could directly check for it, by monitoring the signal at 222 nm (far-UV CD) as a function of temperature, instead of a thermal shift assay which is a more complex way to measure stability changes.

In response to the Reviewer 1's comment, we have plotted the CD signal at 222nm (far-UV) as a function of temperature (newly added Fig S6d), demonstrating that YmoA retained the same secondary structure composition across three different salt concentrations. As mentioned by the reviewer, TSA is indeed a useful tool for assessing thermal stability by measuring the melting temperature of the proteins, and it's more associated with its tertiary structure. By combining the results from these two approaches, we conclude that YmoA remains stable at physiological temperature up to 60°C, under three different salt concentrations (Line 225-227).

Reviewer #2

I appreciate that the authors addressed the points made by this reviewer. However, probably due to a misunderstanding, one major point on transcriptome data still requires a major revision.

Presentation of the transcriptome data is not state of the art and should be revised. First, with few exceptions, the data mentioned in the results section 2.2 is too short, and the figure 2 is too descriptive. The authors should give more details about genes beside T3SS and virulence factor. To this end, a more thorough analysis and much better presentation of the data of figure S2b is required that shows the gene functions grouped according to operons and their function. Much more details about the transcriptome analysis must be mentioned. The whole data set should be provided as supplementary material. What is the significance level, how often were the experiments repeated, why do the authors show only 20 genes most strongly up- or downregulated? Genes involved in virulence or plasmid-encoded should be group. Genes of unknown function should be manually reannotated to provide more functional information.

We sincerely appreciate Reviewer 2's comments and have incorporated the suggestions by extending the results section 2.2 and the Materials and Methods sections. In particular, we have added more detailed information regarding the RNA-seq experiments and data analysis, as well as a newly-added supplementary Table S7. Figure 2a and b, along with FigS2abc and Table S7 illustrate that the experiments were performed with biological triplicates with statistical analysis and listed the most significantly differentially expressed

genes grouped according to fold changes and gene functions. For clarity and focus, we highlighted the top 20 most up-regulated and down-regulated genes to demonstrate the impact of YmoA deletion in Figures. Expanding this list further would detract from the primary focus of our study. It is important to note that section 2.2 represents just one of eight results sections in the manuscript. Extending this section even further would compromise the overall balance and the focus of our study, which is, to elucidate, at the molecular level, how the YmoA protein senses environmental signals and transduces into virulence outputs. Additional analyses are unlikely to yield meaningful insights beyond the current findings.

Reviewer #3

I am satisfied with the revision.